# The Neuropathy Tracker—A mobile health application for ambulatory and self-administred assessment of neuropathy

**Jakob E. Bardram** [1] *, **Mads Westermann**[1‡], **Julia G. Makulec**[1‡], **Martin Ballegaard**[2,3]

**1** Department of Health Technology, Technical University of Denmark, Denmark, **2** Zealand University Hospital, Department of Neurology, Roskilde, Denmark, **3** Department of Clinical Medicine, University of Copenhagen, Denmark

‡ This work is based on the MSc Thesis of these authors [1].
* jakba@dtu.dk

**Data Availability Statement:** Data is available at https://doi.org/10.11583/DTU.27179595.

**Funding:** This work is partly funded by the Copenhagen Center for Health Technology

## Abstract

Peripheral neuropathy is a common neurological disease and is a common complication of diabetes or cancer treatment. Early detection and treatment are crucial for improving the treatment of e.g., diabetic foot ulcers. However, neuropathy detection and monitoring requires examination of the motor and sensory systems and needs to be carried out in a clinical setting by trained professionals, leading to waiting time and delayed treatment. This paper presents the Neuropathy Tracker which is a Mobile Health (mHealth) tool for ambulatory self-assessment of neuropathy, which can be done by the patient at home. The app was designed in an iterative user-centered design (UCD) process involving neurologists, patients, and healthy subjects, thereby ensuring a high degree of clinical validity and usability. The assessment methodology in the app applies state-of-the-art neuropathy assessment methods and the app embodies a user-friendly and systematic assessment flow that guides the patient through the self-assessment. The Neuropathy Tracker tool was subject to a small feasibility study (N = 17), which showed a statistically significant (Pearson correlation $\rho = 0.86$, $p < 0.05$) but moderate (Concordance Correlation Coefficient ($\rho_c$) = 0.69) concurrent validity when compared with a standard clinical assessment method. All patients were able to complete the self-assessment without any help. As such, the technical and user experience design of the Neuropathy Tracker presents a stable mHealth tool that may be feasible for ambulatory self-assessment of neuropathy. Further clinical validation studies are, however, warranted before it is used in the clinical treatment of neuropathy.

## Author summary

Neuropathy is a common complication of diabetes and chemotherapy and if not diagnosed and treated early, it can lead to severe health problems like diabetic foot ulcers. Therefore, early detection, monitoring, and treatment of neuropathy is crucial to most patients. Recently, Mobile Health (mHealth) technologies have increasingly been utilized in healthcare as a novel platform for ambulatory monitoring and treatment by patients

(CACHET) and was done as part of the Brain and Technology Clinical Academic Group (BAT CAG) funded by the Greater Copenhagen Health Science Partners (GCHSP). The funders had no role in study design, data collection and analysis, decision to publish, or preparation of the manuscript.

**Competing interests:** The authors have declared that no competing interests exist.

themselves. In this study, we present a mHealth app called the Neuropathy Tracker for ambulatory self-assessment of neuropathy. The app was designed in close collaboration with neurologists and patients to ensure both a high degree of usability and usefulness as part of the clinical assessment of neuropathy. A small study showed that patients were able to perform the self-assessment procedure on the smartphone themselves and that the results obtained in this process seem to be clinically valid. Further clinical studies are, however, needed to confirm this. In conclusion, this study demonstrates the feasibility of using mHealth technology for ambulatory self-assessment of neuropathy in a neurology patient population.

## Introduction

Peripheral neuropathies are disorders of peripheral nerves located in the extremities. They can be symmetrical across the body axis, asymmetrical, or localized. Patients typically experience numbness, tingling, decreased motor function, pain, or a mixture.

The most common sub-type of peripheral neuropathy is Distal Symmetrical Axonal Poly-neuropathy (DSP). There are multiple causes of DSP, including chemotherapeutic agents, heavy metal exposure, chronic alcohol use, and most prominent; diabetes. Diabetic neuropathy is nerve damage associated with diabetes mellitus [2] and is the most frequent manifestation of diabetes mellitus in the nervous system [3]. In the United States, conservative estimates reckon that DSP affects over 20 million people, more than 10% of the population [4]. Symptoms are present in a distal-to-proximal gradient pattern, meaning severity increases with distance from the body center. Usually, they will reach knee level before appearing in hands [4–6]. Early detection of DSP is crucial for improving morbidity and mortality, such as preventing diabetic foot ulcers [7–9].

There is a variety of measures used by clinicians in DSP diagnosis [4]. They vary in procedures used and the areas of examination. The work-up aims to increase the certainty of a diagnosis from a 'Possible DSP', based on symptoms only, over 'Probable DSP', based on symptoms and clinical examination, to a 'Definite DSP', with added paraclinical tests. To reach a diagnosis of 'Definite DSP', many patients are subject to a Nerve Conduction Study (NCS) [10, 11]. The NCS examines the peripheral nervous system by assessing large myelinated nerve fiber function [12]. This examination is, however, very complex and consists of several parts, requiring a trained professional and specialized clinical equipment. Wait time from referral to a neurology consultation can take multiple weeks, and on average, patients with a suspicion of a peripheral nerve disorder wait longer for their appointment [13]. As an alternative to this highly instrumentalized examination, a set of clinical scoring scales is used in the clinic. These scales do not support a 'Definite DSP' diagnosis, but they increase the diagnostic certainty of a 'Probable DSP', thereby enabling their use in both referral decisions and clinical follow-up. The Utah Early Neuropathy Scale (UENS) is a physical examination scale used for detecting and quantifying early signs of Small Fiber Neuropathy (SFN) in Diabetic Peripheral Neuropathy (DPN) [4, 14]. The examination requires a safety pin, a 128 Hz tuning fork, and a reflex hammer and consists of five types of examination. Motor examination is measured in both legs, and the safety pin is used for measuring the distribution of changes in pin-prick sensation while large fiber sensation is examined through vibration with the tuning fork. UENS has shown a 94% inter-rater reliability, and high specificity and sensitivity. However, no threshold recommended for diagnosis is provided [14]. Total Neuropathy Score (TNS) is another neuropathy scale testing both the legs and arms [4]. It focuses on DPN and Chemotherapy-

Induced Peripheral Neuropathy (CIPN) neuropathy types, offering a broad testing area not focused on a particular phenotype. This scale assesses sensory and motor symptoms, pin-prick and vibration sensation, deep tendon reflexes, and nerve conduction. Both the modified Toronto Clinical Neuropathy Score (mTCNS) [15] and the Douleur Neuropathique 4 Questions (DN4) questionnaire [16] consist of two parts; symptom scores and sensory test scores. The sensory scores are similar to the assessment done in UENS and TNS, but for the symptoms score the patient is additionally asked how much their well-being and daily life are affected by symptoms in their legs, like foot pain, numbness, tingling, weakness, and reduced muscle control. Overall, all of these methods for assessing neuropathy are done in the clinic and require a clinician to perform the tests and issue and score the scales and questionnaires.

Mobile and wearable technology including smartphones has emerged as a promising platform for creating mHealth applications targeting different health domains [17, 18], with a few examples addressing neuropathy [19–22]. These applications are mainly designed to be used by clinicians in the clinic or to collect qualitative survey data from patients. However, by utilizing the participants' own mobile and wearable devices, mHealth technology allows for the longitudinal collection of ambulatory data from study participants, enabling ecologically valid assessment and timely interventions as part of the everyday life of the patient, outside the clinic. This kind of mHealth technology that integrates objective assessment and patient-reported data reported at home could significantly improve the assessment and treatment of neuropathy with minimal inconvenience to the patient and cost to the healthcare system.

This paper presents the design, implementation, and evaluation of the Neuropathy Tracker, which is a mHealth tool enabling ambulatory self-assessment of neuropathy by patients at home. The Neuropathy Tracker was designed in a user-centered design process, involving a set of patients and clinicians, which ensures a high degree of usability and hence potential adoption by patients without the need for trained clinicians to be involved during ongoing self-assessment. The Neuropathy Tracker has been implemented in a cross-platform framework, making the application available on both Android and iOS smartphones. In combination, this design methodology helps ensure broad adoption by end-users (patients) who use their own smartphones. The paper also reports on a small feasibility study. The objective of this study was to assess the feasibility of patients performing a self-administered neuropathy test using the Neuropathy Tracker and to investigate the concurrent correlation of the scores obtained using the app as compared to the golden standard assessment of neuropathy, as scored by a clinician.

## Methods and material

This single-center, prospective design study involved clinicians and patients at the Zealand University Hospital (ZUH) in Roskilde, Denmark, and healthy subjects recruited outside the hospital. The study consisted of two parts; (i) a user-centered design of the Neuropathy Tracker and (ii) a small study evaluating the feasibility of the app for neuropathy assessment.

The study was reported to the relevant Danish ethics authorities. The Danish Medical Agency responsible for overseeing medical device regulation, reported that they would not classify the Neuropathy Tracker as a medical device (letter dated March 9th, 2023). The Ethics Committee of Region Zealand did not classify the study as a clinical study focusing on examining the health of patients, but rather as a quality study focusing on product development. The study was therefore exempted from ethics approval (Journal ID: EMN-2023-01676 and DOK ID: 750691). We used the Douleur Neuropathique 4 Questions (DN4) questionnaire under copyright from Mapi Research Trust, Lyon, France (contact information https://eprovide.mapi-trust.org).

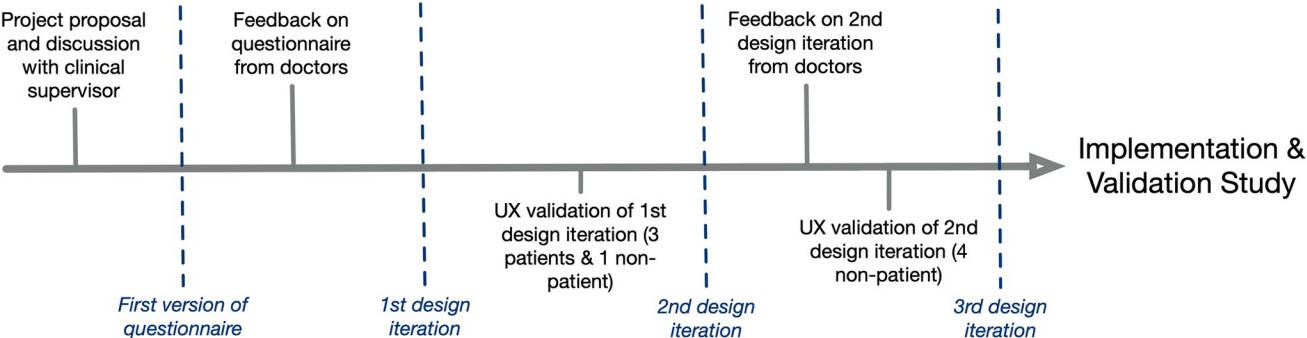

**Fig 1. Design timeline.** Timeline of the user-centered design process of the Neuropathy Tracker illustrating the three major iterations involving clinicians, patients, and healthy subjects ('non-patients').

## User-centered design process

To ensure a high degree of usability and user adoption, the design of the Neuropathy Tracker applied a UCD methodology [23], which has been argued to be essential in the design of mHealth technologies [18]. Two medical doctors (trained neurologists), three neurological patients, and five healthy subjects took part in the design process. The overall timeline of the design process is shown in Fig 1, illustrating that the design of the Neuropathy Tracker was done in three overall iterations, each going through a full design cycle of analysis, design, and validation. Each validation study was video-recorded and important lessons were extracted by video analysis. The lessons from each iteration were incorporated into the next iteration until the final version was ready (presented in the next section). Note that both clinicians, neurological patients, and healthy subjects ('non-patients' in Fig 1) were engaged in the design since all three types of users were considered end-users of the tool; clinicians would need the scores from the tool in their clinical assessment and diagnosis; the neurological patients would use the tool on their own at home; and healthy subjects would either be people at risk (e.g. diabetics and cancer patients) or caretakers for patients.

## Evaluation methods

The feasibility study was designed to investigate the concurrent validity of the Neuropathy Tracker compared to a clinical neuropathy assessment, while also investigating the usability of the app. Concurrent validity shows the extent of the agreement between two assessments taken at the same time, thereby comparing the new assessment method with one that has already been tested and proven to be valid [24]. The predictive power of the new method is analyzed using correlation and linear regression.

The feasibility study was conducted in June 2023 at the outpatient clinic at the Neurological Department of ZUH. Patients at least 18 years old were approached in the waiting room while waiting for an appointment at the hospital. Exclusion criteria were major cognitive impairment and inability to understand and read Danish. When deciding which patients to approach, patients waiting for a neuropathy-related consultation were prioritized more than patients with a different kind of appointment. When first approached, the patients were given a brief explanation of the study and asked if they would like to meet after their appointment for a more detailed explanation. Patients who consented after the detailed explanation were recruited for the study. All participants signed the informed consent before the study. The informed consent complied with the regulation at ZUH.

The study had two parts. First, the participant was provided with a smartphone with the Neuropathy Tracker pre-installed and a safety pin to carry out the study. The participant performed a self-assessment using the Neuropathy Tracker under the observation of a research assistant. Usability was assessed during the first part of the study, applying participant observation techniques [25], while the participant performed the self-assessment. There was no time limit to the test and the patient was told to spend as much time as needed reading the instructions and performing the tests. For the same reason, the tests were not timed. During the second part, the research assistant performed a clinical assessment using the Total Neuropathy Score clinical (TNSc) [26] method, which is the standard for assessing neuropathy in the clinic at ZUH. The research assistant had prior training in performing the TNSc assessment.

Even though the Neuropathy Tracker embodies another assessment approach than TNSc, the TNSc was used for the feasibility study for two reasons. First, TNSc is the preferred method for the assessment of neuropathy in the clinic and is hence the golden standard to compare to. Second, TNSc is designed to be administrated by a trained clinician or researcher and is hence not applicable for self-assessment done by the patient. As such, TNSc is not suited for embedding into an mHealth app like the Neuropathy Tracker.

The order of the two parts was also deliberate. To gauge if the patient was able to understand the instructions in the app and perform the self-assessment on his or her own, it was important that the patient had not been introduced to the terminology and assessment methods during a TNSc assessment. Therefore, the evaluation of the Neuropathy Tracker was always done before the TNSc assessment. We consider the impact of any bias introduced due to this order as minimal since the TNSc assessment is done by the research assistant and is more objective.

## The Neuropathy Tracker

The user-centered design process focused both on creating a tool that was *clinically valid* by taking input from the clinicians while also having a high degree of *usability and understandability* taking input from patients and users in general. The former goal was addressed by having a strong focus on designing the grading design of the tool, while the latter goal was addressed by focusing on the user experience (UX) design. Finally, the technical implementation had to ensure that it could be deployed on the patients' *own* smartphones, including being available on the many different types of smartphones in terms of hardware and operating systems (OSs) present in the general population.

### Grading design

The final design of the Neuropathy Tracker is based on the input from clinicians to become a valid clinical measurement tool for neuropathy. The foundation of the grading tool is the UENS metric, primarily owing to its emphasis on the SFN and early neuropathy. However, in its base form, the UENS scoring sheet is not easily readable for non-medical persons and it requires specialist tools not commonly found in a household, like a tuning fork. Therefore, some adjustments and additions were made to the assessment in the Neuropathy Tracker compared to the UENS metric.

Table 1 provides an overview of the main sections of the assessment in the Neuropathy Tracker. Overall, the tool contains questions and tasks in two categories; graded and non-graded. Responses in the non-graded part can still have individual scores, though they do not contribute toward the overall examination score. This design was decided in order for the tool to provide more contextual information to the examination result for the benefit of the professional by providing information as to whether the patient is experiencing neuropathic pain or

**Table 1. App sections.** Sections of the Neuropathy Tracker grading tool and their scoring.

| Section | Summary | Steps | Scores |
|---|---|---|---|
| Graded Sections | | | |
| General Symptoms | Symptoms in the legs and their severity | 1 | 0-4 |
| Pin-prick Sensation | Investigation into pinprick sensation in 6 sections in both legs | 12 | 0-24 |
| Allodynia/Hyperalgesia | Painful sensation to prick or touch | 4 | 0-4 |
| Large Fiber sensation | Feeling the vibration in three points in both legs | 6 | 0-6 |
| | Feeling changes in great toe joint's position | 2 | 0-2 |
| Motor Examination | Overcoming pressure on the great toes | 2 | 0-4 |
| Non-Graded Sections | | | |
| Neuropathic Pain | Pain Level | 1 | 1-100 |
| | Painful Neuropathy (DN4) Questionnaire | 4 | 0-10 |
| Comments | Any additional information patient wishes to disclose | 1 | N/A |

has additional comments to share. The assessment in each section involved a set of steps (corresponding to screens in the app), listed in the 'Steps' column. The scores for each section are shown in the 'Scores' column. The maximum points possible in the Neuropathy Tracker is 44, comparable to the 42 points in the UENS. The greater the number of points, the higher the likelihood of indicating neuropathy. The scoring in the sections is mostly consistent with UENS and aims to maintain a similar emphasis on early neuropathy detection. Thus over half of the points are dedicated to investigating SFN in the form of pinprick sensitivity.

**General symptoms.** This questionnaire follows the UENS closely while leaving out the part where a clinician would test the patient's deep tendon reflexes. These reflexes are challenging for the patient to test on his or her own without medical training. Instead, the questionnaire in the Neuropathy Tracker contains a question used in other scales like TNS asking if the patient is experiencing any pain, numbness, tingling, or unexpected sensations in the legs. If the patient answers 'yes' to this question, he or she is asked to further specify which body area the symptoms are present; toes, ankles, knees, or above. To avoid mixing self-reported symptoms and more objective findings, this question has been placed first in the examination. This was due to a concern that if a user was asked such a question after a different section, they might drive their answer by what they recorded there. For example, if they learned that they feel reduced pinprick sensation up to the knee, that could influence their choice in this section.

**Pin-prick sensation.** In this section of the examination, the patient should compare the feeling when pricking six areas in each leg to a reference area. This yields twelve steps to be completed, scoring between 0 to 24 points for this part of the examination. For the patient to examine the pin-prick sensation in their legs at home, a fairly sharp, easy-to-use household item was needed. During the UCD process, several 'pointy' household items were tested and the recommendation became to use a safety pin or needle. The original UENS method assesses pin-prick sensation in six areas (the toes, top of the foot, slightly above the ankle, the middle of the lower leg, slightly below the knee, and slightly above the knee) and provides an image of a leg divided into different sections for the clinician to report the findings. However, during the UCD process, we found that this original UENS image was not very informative to the patient as to which area of the leg should be examined. Therefore, a set of new images was iteratively designed with the users, and the final design is shown in Fig 2 (left). This design more precisely illustrates which area should be examined both on the foot and on the side of the leg, as well as which leg is being examined. The pin-prick assessment of the feet and legs needs to be compared to a healthy part of the body. In the design process, it was decided to use the clavicle for

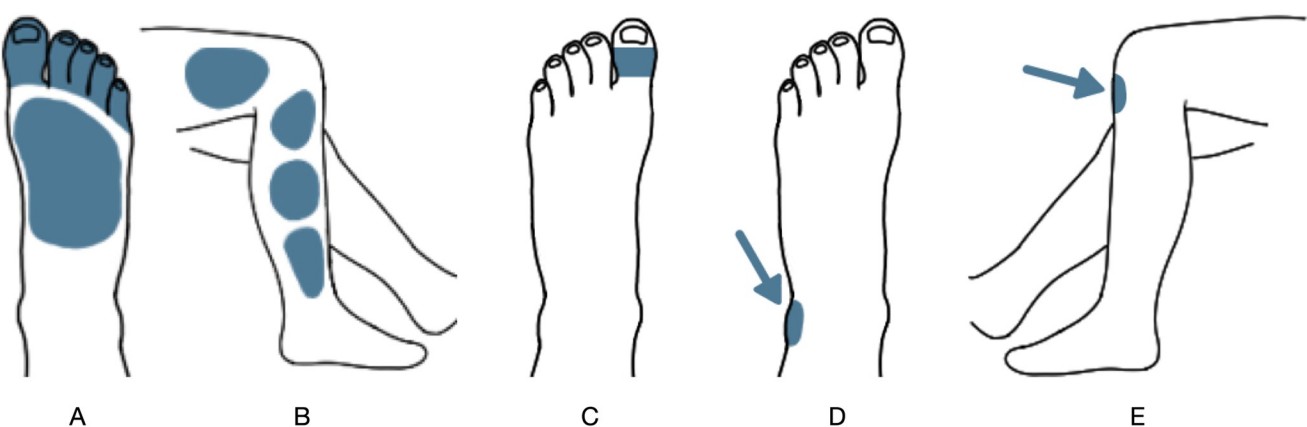

**Fig 2. Pin-prick examination.** The body positions of the pin-prick (A, B) and the large fiber sensation (C, D, E) examination issued to the foot and leg.

comparison. Since DSP affects extremities first this area will be much less affected while being easily accessible for the patient.

**Allodynia and hyperalgesia.** In the original UENS scale, one point is awarded if either allodynia or hyperesthesia are present in the toes or foot of each leg. However, to capture information on which types of hyperesthesia are present, the Neuropathy Tracker issues two separate questions, one about allodynia and one about hyperalgesia. Moreover, in the Neuropathy Tracker, these two questions follow directly from the pin-prick part for each leg reducing the amount of times the patient must switch between legs during the examination.

**Large fiber sensation.** In this section, the patient should examine large fiber sensations by two tests; a vibration test and a joint extension test. In UENS large fiber sensation is only measured in the great toes using a tuning fork and the patient reports if they feel the vibration. During the design of the Neuropathy Tracker, it was decided to use the vibration actuator of a modern smartphone instead of a tuning fork (which is a rare household item). But, depending on the smartphone model, the vibration frequency varies between 130 to 180 Hz [27]. To reduce inconsistencies caused by hardware differences, the Neuropathy Tracker measures vibration only based on its presence, eliminating the 'diminished' option in UENS. However, inspired by the TNS, the Neuropathy Tracker extends the scope of this test to include other areas than the great toes and include vibration measurements on the great toe joint, ankle, and just below the knee cap, as illustrated in Fig 2 (right). The second part of this section examines the great toe's joint position. The users are asked to move their great toe with their hands and answer if they feel the extension in the joint.

**Motor examination.** Following UENS, this section examines motor strength in the great toes by inspecting the dorsiflexion of the toe and determining if the strength of extension is normal or weak. The patient is asked to apply pressure on top of their great toe and try to oppose it by trying to tip up the toe. A score of two is assigned if it is difficult for the toe to overcome the pressure.

**Neuropathic pain.** As part of the design process, it was decided to include a survey on neuropathic pain to the Neuropathy Tracker. This would provide valuable patient-reported information to the clinician. First, the patient is asked to indicate his or her pain level on a scale from 0 to 100 using a visual analog scale [28]. If any pain is reported, the patient is asked to complete a questionnaire based on the DN4 questionnaire [16]. For the patient to self-administer this questionnaire, detailed instructions for some questions were added. For

example, one of the questions in the DN4 questionnaire asks whether pain is provoked or increased by brushing. For the patient to answer this question on his or her own, the Neuropathy Tracker instructs the patient to use his or her fingers to gently stroke the area affected by the pain and then answer if that provoked or increased the pain. As illustrated in Table 1, the scores in this section are not included in the DSP grading score. However, the Neuropathy Tracker keeps the results of this DN4 questionnaire and the pain assessment as data stored in the tool.

## User experience design

An overview of the final UX design of the Neuropathy Tracker is shown in Fig 3. The app consists of four main parts; (i) the home page listing all completed examinations, (ii) the examination pages containing the sections presented in Table 1, (iii) the examination result pages showing the results for each section and the overall neuropathy score, and (iv) the settings page where the patient can add personal information like sex and age, and set preferences like language and vibration duration.

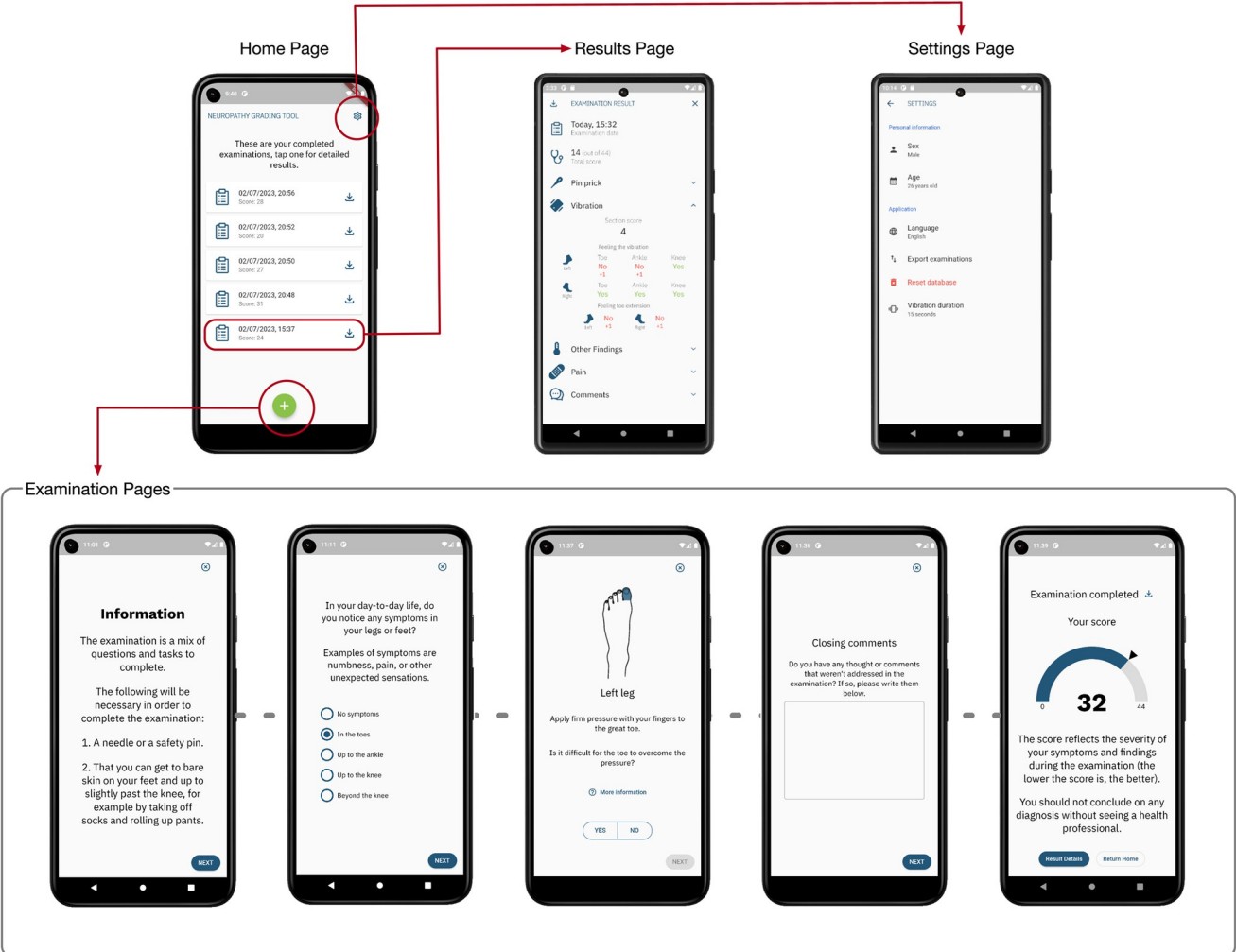

**Fig 3. UX design.** The overall UX design of the Neuropathy Tracker including the main pages and the interaction flow between them. Note that the examination flow comprises 39 pages in total, finishing with the 'Examination completed' page.

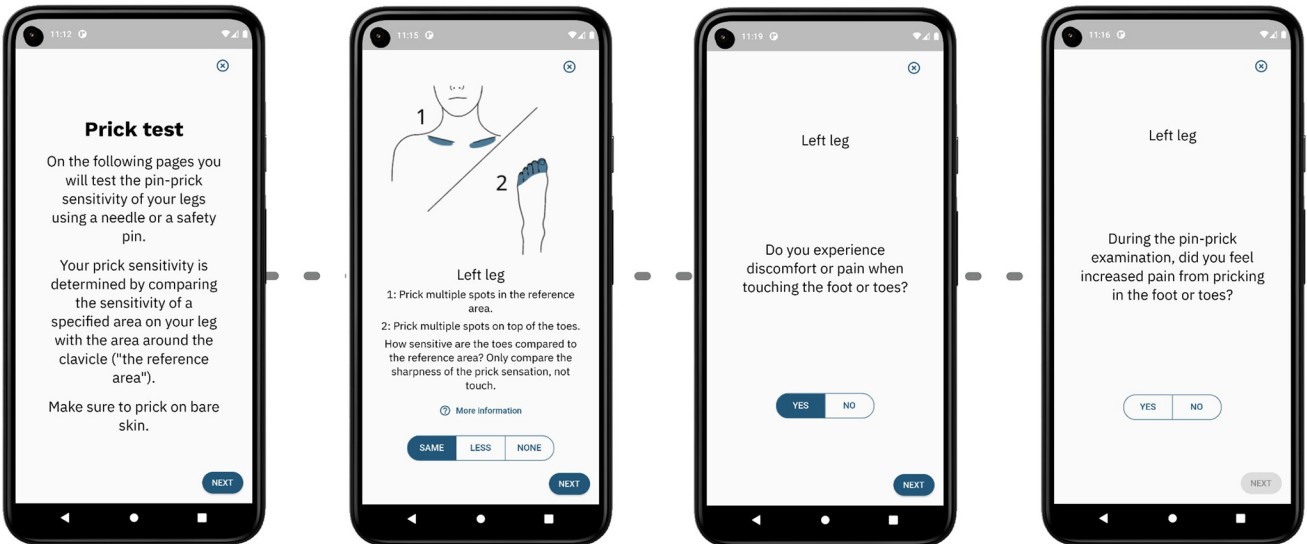

**Fig 4. Pin-prick test UX design.** The UX design of the pin-prick tests showing (i) the instruction page, (ii) the pin-prick test page, (iii) the allodynia test page, and (iv) the hyperalgesia test page. Note that the pin-prick test pages are repeated for all six leg areas and all test pages are repeated for both legs. The illustration on the test pages is adjusted accordingly.

An essential point of the design was developing step-by-step instructions to provide accessible and understandable tasks that patients could conduct themselves. Therefore, each section listed in Table 1 is preceded with detailed instructions on how to prepare and conduct each part of the assessment. As a result, the neuropathy assessment comprises 39 steps in total (34 if no pain is reported), with each step considered a separate page in the Neuropathy Tracker. The flow follows the order listed in Table 1 and ends with the *'Examination Completed'* page, shown at the bottom right in Fig 3. The assessment flow is deliberately designed so that the patient cannot go back to previous steps or proceed to the next step before the current step is done. This ensures that tests are not altered once done and that the patient completes all tests. Many steps are repetitive, where the instructions, tests, and questions are similar, but target a different leg or body area.

Fig 4 shows the flow of the pin-prick test section. Following the general description of the test (first page), it provides instructions for each section of the leg for both the left and right leg. On each page, the illustration is adjusted to show what leg area and side are to be tested. More detailed instructions and description can be obtained by pressing the *'More Information'* button, including a detailed explanation of what is meant by the *'same'*, *'less'*, and *'none'* answers. The pin-prick tests are immediately followed by the allodynia and hyperalgesia tests, one for each leg.

Fig 5 shows the flow of the vibration and motor tests. Following the general description of the test (first page), it provides instructions for performing the vibration test in different areas of the feet and legs, and for both sides. All illustrations are adjusted accordingly. In each vibration test, the patient is asked to press the back of the phone firmly against the body part being examined, like the *"Top side of the bone in your great toe"*, *"Outside part of your ankle"*, or *"Bone just below your kneecap"*. When the user presses the start button, the phone vibrates, and the button changes color to red, displaying *"Stop"*, allowing the patient to stop the vibration. In the great toe joint position test (Fig 5(iii)) the patient is asked to put the phone down since it would be difficult to perform the test while holding the phone. For the

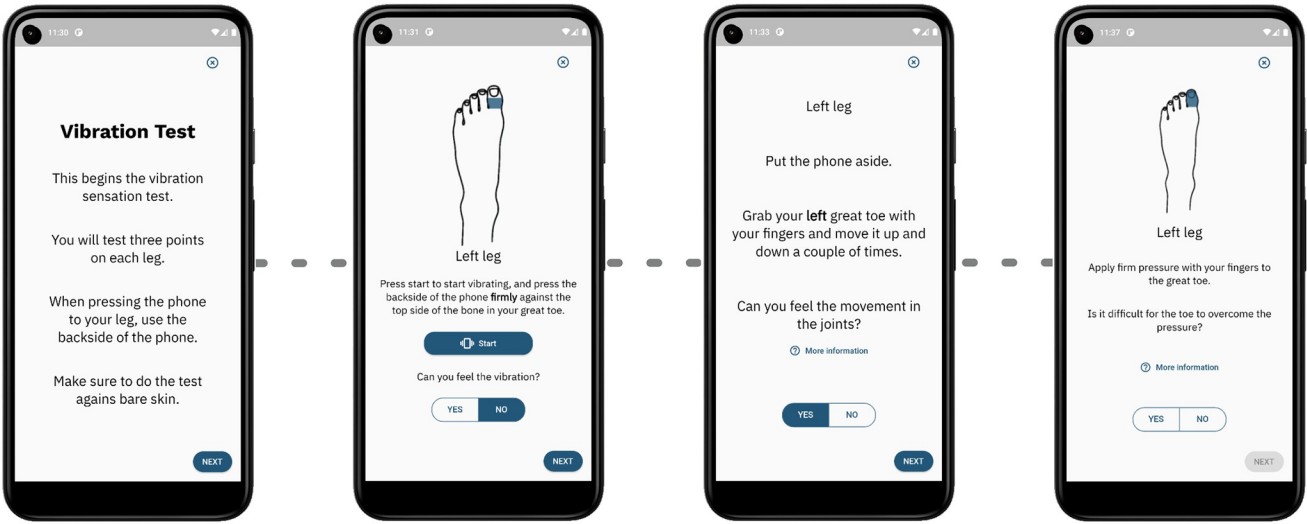

**Fig 5. Vibration and motor test UX design.** The UX design of the vibration and motor tests showing (i) the instruction page, (ii) the vibration test page, (iii) the great joint test page, and (iv) the motor test page. The test pages are repeated for each leg and the illustration is adjusted accordingly. Each test has its own instruction page (not shown).

motor examination step (Fig 5(iv)), the patient is asked to apply pressure on the great toe on both feet. On all test pages, the *'More Information'* button will show a detailed description for the patient on how to perform the test and how to gauge their sensation. For example, in the motor test, the information page describes how the patient can compare how well their fingers overcome the same applied pressure, and answer *'yes'* if their great toe seems weaker than that.

Fig 6 shows the flow of the pain section. If the patient feels pain, s/he is asked to indicate the pain level using a slider with a scale from 0 to 100 (Fig 6(ii)). Then the patient is asked to fill in a set of questions based on the DN4 questionnaire [16], each investigating the patient's pain characteristics and symptoms reported using multiple-choice questions (Fig 6(iii) and 6(iv)).

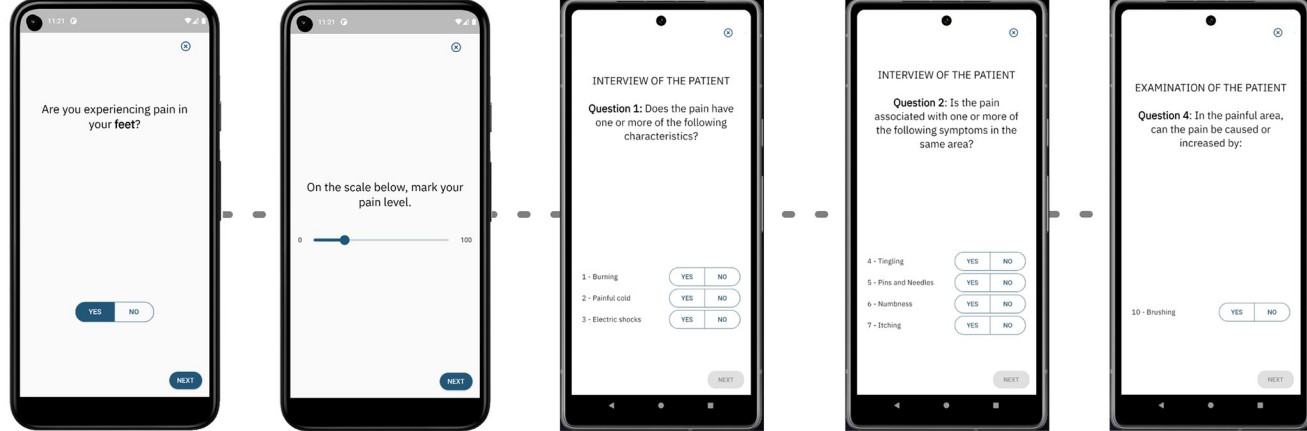

**Fig 6. Pain test UX design.** The UX design of the pain test showing (i) the question on pain precense, (ii) pain level, (iii) DN4 pain characteristics, (iv) DN4 pain symptoms, and (v) DN4 brushing response. Note that the DN4 questions are sample copies, do not use them without permission.

The section ends with the brushing test asking the patient to brush him or herself in the painful area and gauge if this increases or provokes pain in that body area.

When the patient has performed all the sections and tests, the result of the examination is shown on the *'Examination Completed'* page (Fig 3 bottom right), which shows the overall examination score on a gauge from 0 to 44 accompanied with a text explaining that this scores should be interpreted only by a health professional. From this page, the patient can download the detailed test results, navigate to the result page to view the data or navigate back to the home page.

The *'Results'* page provides an overview of the results from an examination, showing the overall score and the scores for each sub-section of the assessment. Each section is shown as a fold-down card with detailed images and scores for the different tests. For example, in Fig 3, results from the vibration test are shown as the overall score for this section in combination with detailed answers using icons for the left or right foot and color-coding in red or green according to the patient's answer. The results of other sections use other icons and illustrations that resemble the test and answers. For example, the pain result section displays the pain slider and the DN4 questionnaire responses, each illustrated with an icon that symbolizes the answer, such as a 'fire' icon to resemble the answer that *"pain feels like burning"*.

### Technical implementation

The Neuropathy Tracker is implemented using the Flutter user interface (UI) framework version 3. Flutter is a cross-platform framework that enables the implementation of smartphone apps, which compiles natively to both the Android and iOS operating systems (OSs). This ensures a consistent User experience across all smartphones that patients might own.

Fig 7 shows the overall software architecture of the Neuropathy Tracker. It consists of a set of UI components (marked red) as shown in Figs 3 to 6 and presented above, and a set of business logic components (green). For implementing the core logic of the assessment flow, the Neuropathy Tracker relies on the Research Package (RP) [29] framework (orange components). Research Package (RP) allows for implementing the different steps and linking these together in the overall assessment flow, including skipping parts of the assessment (in the case of Neuropathy Tracker skipping the neuropathic pain section if there is no pain reported). To implement the specialized tests and questions in the Neuropathy Tracker, a set of neuropathy-specific question steps was created as an extension to RP. For example, a custom `PainSliderQuestionStep` showing the slider for pain level (see Fig 6) and a `VibrationQuestionStep` which can start and stop the vibration on the phone (see Fig 5). RP is also responsible for the localization (i.e., translation) of the app, which is available in English and Danish. Finally, the Neuropathy Tracker makes use of a set of Flutter plugins, the main ones shown in Fig 7 (marked blue), which again access operating system-specific services (yellow) like the file system and the vibration actuator.

The Neuropathy Tracker stores data locally in JSON on the phone using a NoSQL database and data can be exported in a comma-separated values (CSV) format. The patient's settings and preferences are similarly stored locally.

### Results

The feasibility study aims to investigate if the Neuropathy Tracker self-assessment tool can be useful in detecting DSP and assessing its severity. This is done by examining two parameters; (i) concurrent validity between the assessment obtained from the Neuropathy Tracker and the reference standard Total Neuropathy Score clinical (TNSc) assessment, and (ii) the qualitative usability of Neuropathy Tracker when used by the patient for neuropathy self-assessment.

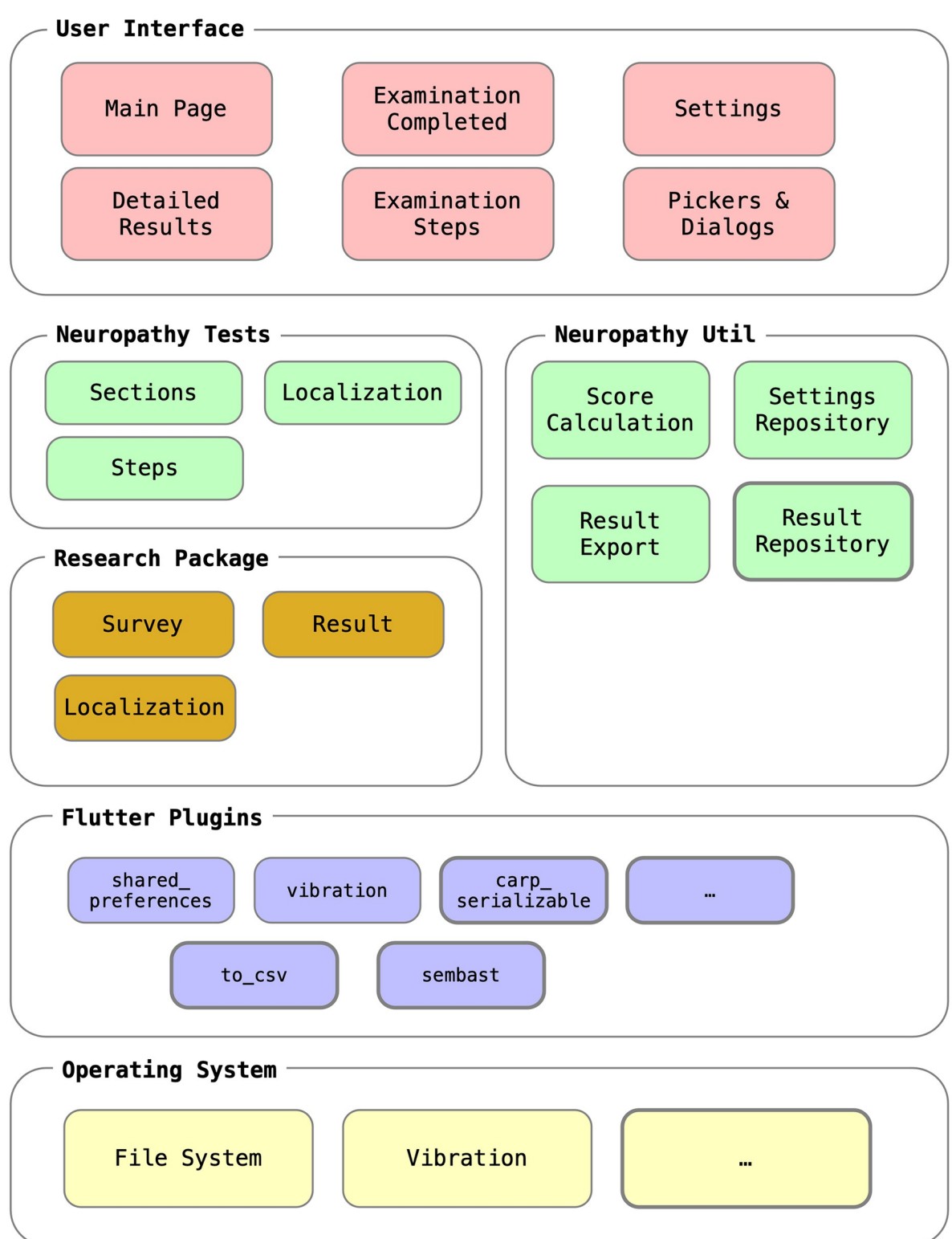

**Fig 7. Software architecture.** Software architecture diagram of the Neuropathy Tracker comprising of (i) UI components (red) as shown in Figs 3 to 6; (ii) business logic components (green); (iii) Research Package (RP) components (orange); (iv) Flutter plugins (blue), and (v) the phone's operating system services (yellow).

## Participant characteristics

All patients were recruited in the outpatient clinic at the Department of Neurology at ZUH. Recruitment was done from all neurological patients visiting the outpatient clinic, including, but not limited to, neuropathy patients, who had very different neuropathy trajectories with some newly diagnosed and others with longstanding neuropathy problems. This recruitment strategy was deliberately designed to include as many different types of patients with different levels of experience with neuropathy. The inclusion of a broad spectrum of patients would help establish concurrent validity on both ends of the neuropathy scale, as well as gather usability input from a heterogeneous group of patients.

A total of 28 patients were invited to the study. Of these, 10 patients declined to participate and one patient did not finish the study since her taxi arrived earlier than expected. Hence, 17 patients concluded the study, 5 being male and 12 female (70%). Ages ranged from 40 to 75, median age of 55 years. Table 2 provides an overview of the recruited patients and the dataset is publicly available [30]. All patients showed some signs of neuropathy based on the TNSc assessment. For privacy reasons (c.f., the ethical exemption), no sensitive information like names, social security numbers, or diagnoses were recorded. Both the self-assessment and the clinical assessment were completed for each participant during the same session, in this order. The same research assistant was responsible for both parts of the test.

## Validation procedure

The validation study took place in an examination room at the outpatient clinic at ZUH with assessment sessions completed over 6 days (c.f. Table 2).

**Part I—Self-assessment.** In the first part, the patient performed the entire self-assessment using the Neuropathy Tracker. Patients were provided with an Android smartphone with the app installed and a safety pin. All patients used the Danish version of the app. The patient was asked to follow the instructions in the app and perform the self-assessment, as instructed. The

**Table 2. Participants.** Study participant characteristics and neuropathy scores obtained from the Total Neuropathy Score clinical (TNSc) and Neuropathy Tracker assessment.

| Date | Sex | Age | TNSc | Neuropathy Tracker |
|------|-----|-----|------|--------------------|
| 05.06.2023 | Female | 42 | 2 | 7 |
| 05.06.2023 | Male | 41 | 15 | 19 |
| 05.06.2023 | Female | 40 | 1 | 2 |
| 06.06.2023 | Female | 55 | 16 | 22 |
| 06.06.2023 | Female | 48 | 19 | 37 |
| 07.06.2023 | Female | 55 | 4 | 2 |
| 07.06.2023 | Female | 48 | 6 | 5 |
| 07.06.2023 | Male | 67 | 10 | 25 |
| 07.06.2023 | Female | 75 | 13 | 25 |
| 08.06.2023 | Male | 58 | 24 | 26 |
| 08.06.2023 | Female | 73 | 10 | 6 |
| 08.06.2023 | Male | 65 | 7 | 0 |
| 08.06.2023 | Female | 51 | 4 | 7 |
| 09.06.2023 | Female | 63 | 16 | 20 |
| 09.06.2023 | Male | 40 | 13 | 13 |
| 13.06.2023 | Female | 58 | 24 | 37 |
| 13.06.2023 | Female | 52 | 9 | 12 |

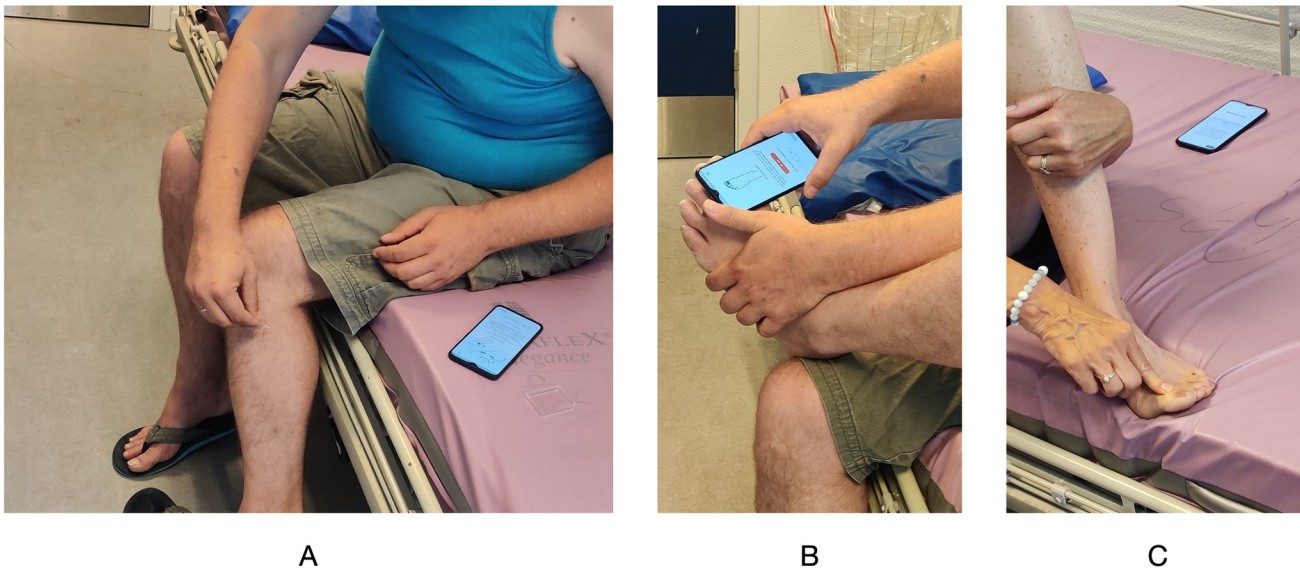

**Fig 8. Study part I.** A study participant carrying out the self-assessment performing the (A) pin-prick, (B) vibration, and (C) motor tests.

research assistant remained in the room and took usability notes, but did not in any way interfere with the patient's self-assessment. Fig 8 shows a patient performing some tests using the app.

**Part II—Total Neuropathy Score clinical (TNSc).**  In the second part, the TNSc assessment was done by the research assistant, who had been trained in performing this assessment. Fig 9 shows some of the tests done as part of the TNSc assessment. In order to align with the test done in the Neuropathy Tracker, TNSc was issued on the legs only. The assessment followed the TNSc method, asking the patient about the four sensory symptoms, sensations in the body areas (feet, legs, hand, arms), and motor symptoms. The pin-prick sensation was done using the same type of safety pin as in the self-assessment part and issued on the feet and legs, using the clavicle as a reference area. A 128 Hz tuning fork was used for vibration sensation testing in the leg, applying it to all four bones (the bone on top of the great toe, the outer ankle bone, the bone right below the knee, and the hip bone). For motor strength, the ankles, fingers, and wrists were all tested, but reflexes were only tested in the ankles with the Achilles tendon and in the knee with the patellar tendon. Finally, patients were asked about autonomic

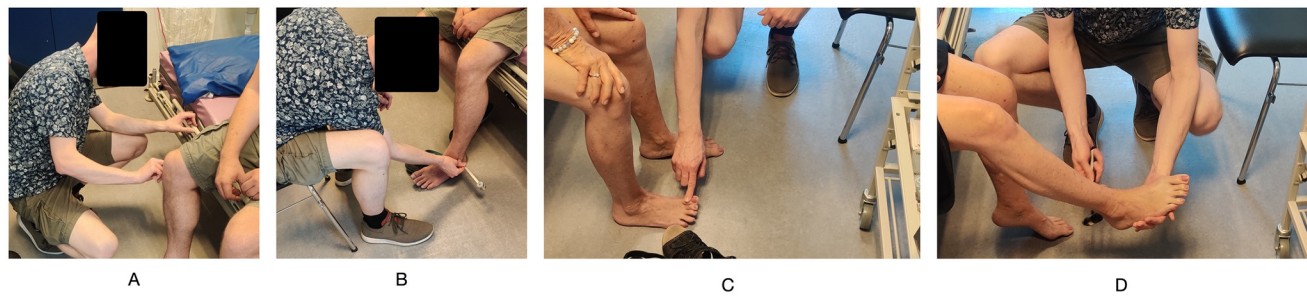

**Fig 9. Study part II.** The research assistant carrying out the Total Neuropathy Score clinical (TNSc) assessment on a study participant performing the (A) pin-prick, (B) vibration, (C) motor, and (D) reflex tests.

symptoms according to the TNSc questionnaire (loss of bowel and bladder control, fainting, impotence, or constipation).

## Concurrent validity

Table 2 shows the results from the study for each patient, including the date of assessment, the biological sex and age of the patient, and the overall TNSc and Neuropathy Tracker assessment scores. When comparing the TNSc and app scores, the Pearson Correlation Coefficient ($\rho$) is 0.86 with a p-value of 0.0000188, i.e., less than 0.05. Hence, a statistically significant correlation is found. Fig 10 shows a Bland-Altman plot with the differences between the Neuropathy Tracker ("App") and TNSc measures on the y-axis and their means on the x-axis. The plot reveals that there is an agreement between the app and the TNSc scores. All data points except one are within the limits of agreement (LoA) and 11 out of 17 data points are within the mean ±1 SD. We do, however, observe a small proportional bias where the app scores lower than TNSc for lower scores and higher for higher scores. To further investigate the agreement of the two scores, a $\rho_c$ is calculated. The $\rho_c$ is designed specifically to measure agreement rather than just association and evaluates how far each pair of measurements deviates from the line of perfect agreement (the 45° line in a Pearson correlation). For the data set, the overall $\rho_c$ value is 0.69, which indicates a moderate agreement. Hence, while there is a reasonable level of consistency between the TNSc and 'App' measures, there are some discrepancies that should be considered.

## Usability

All study participants could perform the self-assessment using the Neuropathy Tracker without any help from the observing research assistant, and no blocking usability issues were raised or found during part I. Two issues were, however, raised by some patients. First, a few patients

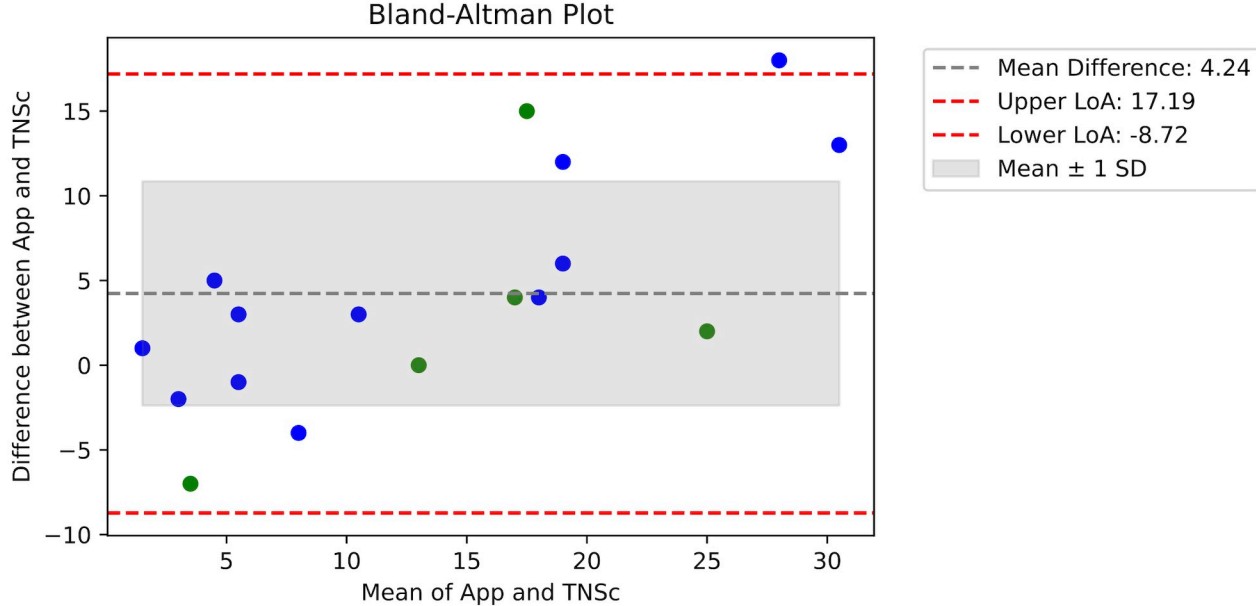

**Fig 10. Concurrent validity.** A Bland-Altman plot with the differences between the Neuropathy Tracker ("App") and TNSc measures on the y-axis and their means on the x-axis. The plot includes horizontal lines for the mean difference and the limits of agreement (LoA) and is color-coded by sex (green = male, blue = female).

were left-right confused and could not always immediately understand which foot or leg to perform the test on. The pictures included in the app (see Fig 2) helped them choose the right side and no mistakes were observed during the study. Second, all patients used the default vibration level as set in the user settings. However, some patients mentioned that it was sometimes difficult to differentiate between the vibration sensing in the hand holding the phone and the area of the foot or leg that the phone was held against.

## Discussion

### Principal findings

By applying a user-centered design methodology involving two clinicians, three patients, and five healthy subjects, the Neuropathy Tracker was designed and implemented. The Neuropathy Tracker presents a novel mHealth tool for ambulatory and continuous self-assessment of neuropathy by the patient him- or herself. To maintain high clinical validity, the tool embodies a unique combination of established assessment methods for neuropathy, including the UENS, TNS, and DN4 scoring systems. To ensure high usability and understandability for the patient while doing these rather complex assessment tasks on his or her own, a thorough effort was put into the UX design of the Neuropathy Tracker. To ensure that patients can run the Neuropathy Tracker on their own phones, the app has been implemented in the cross-platform Flutter framework, thereby ensuring a consistent UI and availability across the many different smartphone models and platforms available. Moreover, the app relies only on common household items for its tests (like a safety pin) while using the vibration actuator on the phone for vibration tests.

The Neuropathy Tracker was subject to a small feasibility study involving N = 17 patients who were recruited in a neurological outpatient clinic with a varying degree of neuropathy. The study showed a statistically significant but moderate agreement between the overall scores obtained by the self-assessment done by patients using the Neuropathy Tracker and the overall scores obtained by the Total Neuropathy Score clinical (TNSc) assessment. In terms of usability, all participants were able to perform the self-assessment relying solely on the instructions and UX design of the app. However, some usability issues especially related to the configuration and use of the vibrator on the phone were found. But overall, the Neuropathy Tracker seems feasible for self-assessment of neuropathy both from a clinical and patient perspective.

### Comparison with prior work

A few mHealth applications for neuropathy have been presented in the literature. Piaggio et al. [19] present an Android-based mHealth app for screening diabetic neuropathies in rural Africa. The app is designed to walk a clinician through three tests; the Neuropathy Total Symptom Score-6 (NTSS-6) survey, a two-point discrimination pin-prick test, and a vibration perception test. The latter two tests are supported by two 3D-printed devices for issuing the pin-prick and for mounting on the phone to enhance the phone's vibrator. Similar to the Neuropathy Tracker, this app also uses the vibration actuator on the phone. However, it is not designed for self-assessment by the patient but is designed to be used by the clinician, who would install the app on their phone. Moreover, the app is only available on Andriod, which makes sense in Africa where Android runs on 85% of all smartphones, while in Denmark, more than half the population (58%) uses iPhones. Hence, a self-assessment solution needs to be available across many different platforms and types of smartphones.

Oh & Lee [20] presents an mHealth application for a self-management program for Chemotherapy-Induced Peripheral Neuropathy (CIPN) patients. It uses questionnaires to assess CIPN's severity, negative impact on daily life, and quality of life. The app also works as a

learning tool about CIPN and as a coach, suggesting exercises for relieving symptoms. This app only supports questionnaires and no objective assessment tests, like the Neuropathy Tracker does. Moreover, the paper focuses on assessing the 'value' of the app for the patient in terms of the reduction of symptoms and improvement in quality of life. Thus, the reliability of the self-assessment and its clinical validity and precision have not been evaluated.

The NeuroDetect mHealth app [21] is also aimed at patients with CIPN and is used for assessing the severity of the condition. The application makes an assessment based on two types of functional tests, one testing the strength and dexterity of the hands, and the other testing gait and balance. A partial least-squares discriminant analysis of NeuroDetect results showed a good ability to differentiate between CIPN patients and healthy controls. This app, however, only assesses gait and balance, not sensation, which is core to the assessment of neuropathy. Furthermore, the app is highly dependent on the 'Gait and Balance' and '9-Hole Peg' tests only available within Apple's ResearchKit, thereby making the app only available on iOS.

PeriVib [22] is the combination of an mHealth application and a portable vibration device that enables assessment and diagnosis of CIPN. The vibration device is used for determining the vibration sensitivity threshold of the patient's feet with vibration tests, and the mobile app for performing gait and sway functional tests to assess the presence of CIPN. The vibration threshold test was correlated with a biothesiometer and tuning fork, with coefficients $R^2 =$ 0.68% and $R^2 = 0.15$% respectively. The motor examination—balance and gait tests—correlated poorly with the threshold test and severity of CIPN. Neither the proprietary vibration device nor the app are publicly available, and no further studies on the tool have been reported. Moreover, this solution is for clinical use and not for self-assessment.

## Impact on patient care

For the subject experiencing symptoms, the Neuropathy Tracker will enable a structured and validated grading of those symptoms and guide the subject through a structured self-evaluation of findings suitable for reaching levels of information comparable to meeting a diagnosis of 'Probable DSP'. As a clinical diagnosis will require evaluation by a certified clinical professional, the Neuropathy Tracker will be optimally placed to guide attention to this health issue on referral for diagnostic visits or to guide the frequency and timing of visits for subjects at risk of developing DSP. A possible setting could be symptom-initiated evaluations in patients under treatment with neurotoxic medication as during chemotherapy for cancer. Between clinical visits, repeated evaluations could help patients and health professionals to evaluate the level of neurotoxicity.

For the health professional wanting to keep informed about DSP development, the Neuropathy Tracker could track signs and symptoms, thereby enabling symptom-initiated virtual clinical visits or a more scheduled series of evaluations. This would be useful for surveillance of neuropathy complications in diabetes, hypovitaminosis after gastric surgery and banding, and as follow-up after lumbar surgery for spinal stenosis.

As such, the Neuropathy Tracker will enable communication about current symptoms and pseudo-objective findings for people with peripheral nerve health problems. However, based on the small feasibility study presented in this paper, it is not possible to identify detection thresholds for clinical conditions thereby suggesting a diagnostic property of the tool. This would warrant larger and more detailed validation studies.

## Limitations

There are also limitations to the study to be acknowledged. From a technical perspective, we found that the vibration actuator behaves differently across smartphone models and OSs, and

from a usability perspective we found that some patients had different levels of sensitivity to the vibration. The use and configuration of the vibration actuator thus warrants a thorough investigation and comparison to the tuning fork if used in a larger population. Additionally, the Neuropathy Tracker was only tested by patients in the Danish Android version and more technical and usability testing might be needed on other smartphone models and languages. The self-assessment of patients was done in an examination room in the outpatient clinic and not at the patient's home. Home-based neuropathy assessment might lead to new usability issues that should be addressed. Moreover, the study included a rather homogeneous group of patients, and including a more heterogeneous patient group with different visual, motor, or cognitive impairments might reveal important accessibility issues.

The feasibility study was conducted in a single clinical setting with a relatively small sample size, affecting the generalizability of study results. The study only investigated the validity of the overall app score as compared to the overall TNSc score. The study did not power this limited validation for anything beyond this. As a result, we can only report the external validation based on the overall TNSc score and cannot make any claims regarding any sub-scores from the sub-sections and how they may correlate with the examination performed in the clinic. Moreover, the study setup might induce bias or errors in the assessments for at least three reasons; (i) the assessment using the Neuropathy Tracker was always performed before the TNSc, (ii) the same research assistant was responsible for both sub-parts of the test, and (iii) the TNSc assessment was done by newly trained non-clinical persons. Finally, the present feasibility study was a 'one-off' test that did not assess the reliability of the Neuropathy Tracker's measurements over time by applying a test-retest validation setup. Based on the experience from this feasibility study, future studies have been designed to better evaluate both the clinical accuracy and validity of the obtained scores, as well as the usability and usefulness of the Neuropathy Tracker when used both in the clinic and the patient's home.

## Conclusion

This paper has presented the design, implementation, and feasibility evaluation of the Neuropathy Tracker, which is an mHealth technology for ambulatory self-assessment of peripheral nerve health problems. The Neuropathy Tracker was designed in a UCD process involving clinicians, patients, and healthy subjects. The assessment methodology in the app is based on state-of-the-art neurological assessment methods thereby ensuring a high degree of clinical alignment. An iterative UX design process involving both neurological patients and health subjects ensured a high degree of usability and the app embodies a UI that systematically guides the patient through the self-assessment process at home. The app was implemented in a cross-platform programming framework making it available across different hardware and operating system versions, and in different language translations. As such, the usability and availability of the app enable widespread adoption of this self-assessment approach.

The Neuropathy Tracker was subject to a small feasibility study involving 17 patients with a varying degree of neuropathy. The study showed a statistically significant (Pearson correlation $\rho = 0.86$, $p < 0.05$) but moderate (Concordance Correlation Coefficient ($\rho_c$) = 0.69) agreement between the overall scores obtained by the self-assessment done by patients using the Neuropathy Tracker and the overall scores obtained by the Total Neuropathy Score clinical (TNSc) assessment. In terms of usability, all participants could perform the self-assessment relying solely on the instructions in the app. In conclusion, the Neuropathy Tracker seems to be feasible for ambulatory self-assessment of neuropathy both from a clinical and patient perspective. The overall goal is to use the Neuropathy Tracker in clinical diagnosis and treatment of

neuropathy as part of clinical treatment in a clinic. However, more thorough clinical validation in larger studies is warranted first.

## Acknowledgments

The authors want to thank Prof. Nanna Brix Finnerup at the Danish Pain Research Center at Aarhus University for providing feedback on the assessment design in the Neuropathy Tracker.

## Author Contributions

**Conceptualization:** Jakob E. Bardram, Mads Westermann, Julia G. Makulec, Martin Ballegaard.

**Data curation:** Mads Westermann, Julia G. Makulec.

**Formal analysis:** Mads Westermann, Julia G. Makulec.

**Funding acquisition:** Jakob E. Bardram, Martin Ballegaard.

**Investigation:** Mads Westermann, Julia G. Makulec.

**Methodology:** Jakob E. Bardram, Mads Westermann, Julia G. Makulec, Martin Ballegaard.

**Project administration:** Jakob E. Bardram, Martin Ballegaard.

**Resources:** Jakob E. Bardram, Martin Ballegaard.

**Software:** Jakob E. Bardram, Mads Westermann, Julia G. Makulec.

**Supervision:** Jakob E. Bardram, Martin Ballegaard.

**Validation:** Mads Westermann, Julia G. Makulec, Martin Ballegaard.

**Visualization:** Jakob E. Bardram, Mads Westermann, Julia G. Makulec.

**Writing – original draft:** Jakob E. Bardram.

**Writing – review & editing:** Jakob E. Bardram, Mads Westermann, Julia G. Makulec, Martin Ballegaard.

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
