## [Decision Letter · Decision Letter 0]

30 Sep 2024

PDIG-D-24-00371

The Neuropathy Tracker – A mobile health application for ambulatory and self-administred assessment of neuropathy

PLOS Digital Health

Dear Dr. Bardram,

Thank you for submitting your manuscript to PLOS Digital Health. After careful consideration, we feel that it has merit but does not fully meet PLOS Digital Health's publication criteria as it currently stands. Therefore, we invite you to submit a revised version of the manuscript that addresses the points raised during the review process.

Please submit your revised manuscript within 60 days Nov 29 2024 11:59PM. If you will need more time than this to complete your revisions, please reply to this message or contact the journal office at digitalhealth@plos.org. Please include the following items when submitting your revised manuscript:

We look forward to receiving your revised manuscript.

Kind regards,

Haleh Ayatollahi

Section Editor

PLOS Digital Health

Journal Requirements:

1. We have amended your Competing Interest statement to comply with journal style. We kindly ask that you double check the statement and let us know if anything is incorrect. 

2. Your current Financial Disclosure states, “The author(s) received no specific funding for this work.”. However, your funding information on the submission form indicates that you received funding from “Copenhagen Center for Health Technology" with Grant Recipient "Prof. Jakob E. Bardram" and "Greater Copenhagen Health Science Partners" with Grant Recipient "Dr. Martin Ballegaard" and Grant Number " BAT-CAG". Please indicate by return email the full and correct funding information for your study and confirm the order in which funding contributions should appear. Please be sure to indicate whether the funders played any role in the study design, data collection and analysis, decision to publish, or preparation of the manuscript.

3. We ask that a manuscript source file is provided at Revision. Please upload your manuscript file as a .doc, .docx, .rtf or .tex.

4. We noticed that you used “not shown” in the manuscript. We do not allow these references, as the PLOS data access policy requires that all data be either published with the manuscript or made available in a publicly accessible database. Please amend the supplementary material to include the referenced data or remove the references.

5. Figure 10 includes an image of an identifiable person. Please provide written confirmation or release forms, signed by the subject(s) (or their guardian), giving permission to be photographed and to have their images published under a Creative Commons license. You may upload permission forms to your submission file inventory as item type 'Other'. Otherwise, we kindly request that you remove the photograph.

6. Please upload a copy of Figures 1, 2, 3, 4, 5, 6, 7, 8, 9, 10, and 11 which you refer to in manuscript. Or, if the figure is no longer to be included as part of the submission please remove all reference to it within the text.

7. Thank you for uploading your study's underlying data set. Unfortunately, the repository you have noted in your Data Availability statement does not qualify as an acceptable data repository according to PLOS's standards.

8. Figures 2 and 3: Please confirm whether you drew the images / clip-art within the figure panels by hand. If you did not draw the images, please provide (a) a link to the source of the images or icons and their license / terms of use; or (b) written permission from the copyright holder to publish the images or icons under our CC-BY 4.0 license. Alternatively, you may replace the images with open source alternatives. See these open source resources you may use to replace images / clip-art:

- https://openclipart.org/

9. Figures 4, 5, 6, and 7 contains screenshots. We are not permitted to publish these under our CC-BY 4.0 license; websites are usually intellectual property and are copyrighted.This includes peripheral graphics of the web browser such as the [X] buttons. We ask that you please remove or replace it.

10. Figure 10 includes an image of an identifiable person. Please provide written confirmation or release forms, signed by the subject(s) (or their parent/legally authorized guardian), giving permission to be photographed and to have their images published under our CC-BY 4.0 license. 

Otherwise, we kindly request that you remove the photograph.

Additional Editor Comments (if provided):

Reviewers' comments:

Reviewer's Responses to Questions

**Comments to the Author**

1. Does this manuscript meet PLOS Digital Health’s publication criteria? Is the manuscript technically sound, and do the data support the conclusions? The manuscript must describe methodologically and ethically rigorous research with conclusions that are appropriately drawn based on the data presented.

Reviewer #1: No

Reviewer #2: Yes

2. Has the statistical analysis been performed appropriately and rigorously?

Reviewer #1: No

Reviewer #2: No

3. Have the authors made all data underlying the findings in their manuscript fully available (please refer to the Data Availability Statement at the start of the manuscript PDF file)?

Reviewer #1: No

Reviewer #2: Yes

4. Is the manuscript presented in an intelligible fashion and written in standard English?

Reviewer #1: Yes

Reviewer #2: Yes

5. Review Comments to the Author

Reviewer #1: The manuscript is well-written, clear, and presents an interesting approach to addressing an important issue in neuropathy detection and monitoring. The use and detailed description of methodological approach of the user centered design is a strong characteristic of the paper, as well as the depiction and detailed description of how the app works. However the use of this methodology does not “ensure a high degree of clinical validity and usability” as stated in the manuscript. So does not the feasibility study conducted, given the small number of subjects (N=17) and the selection bias (once all participants were recruited at the same clinic). In order to sustain such a claim a much larger and rigorous validation study would be necessary. A more cautious and tentative language would be advised to describe the results of the study, which is relevant though preliminary. A major problem of the study is the fact that the total neuropathy score clinical, which is a central result, was assessed by only one person who is described to be a newly trained non-clinical research assistant. This limitation compromises the rigor of the validation study,

Reviewer #2: • Overall:

 ○ The authors describe the creation of the "Neuropathy Tracker", a mobile health tool designed for ambulatory neuropathy self-assessment. This represents overall, a reasonable initial feasibility study, although clearly additional study would be needed before releasing this broadly to patients. 

 ○ I commend the authors on the clear degree of detail, as often the papers for mobile health applications do not give a clear sense of their actual contents and function 

 ○ Overall, however, my greatest concern is that the paper has somewhat excessive information in parts of the methods section, as well as a lack of sufficient detail in the area of the results. Some of the intense technical detail would benefit from being moved to a supplement, particularly regarding the computational software strategy (although the detail around the decision-making behind the project is excellent). The results section come on the other hand, has only a general correlation metric while I would like to see a little bit more of the subtest by subtest metrics.

 ○ I have some other concerns specific to methodology:

 ○ With respect to methodology, please consider explaining why TNSc was chosen as the comparison standard rather than UENS, given that the Neuropathy Tracker is more closely based on UENS.

 ○ Discuss potential biases in the study design, such as the order effect of always performing the Neuropathy Tracker assessment before the TNSc.

Specific comments:

 • Abstract:

 ○ The abstract is clear in its overall structure and content.

 ○ Include more specific results from the feasibility study, such as the correlation coefficient and p-value.

 • Author Summary:

 ○ Author summary is accurate and high-quality

 • Introduction:

 ○ I enjoy the degree of detail in the introduction as a neurologist, however I wonder if there may be some excessive detail given about the many different neuropathy scores and questionnaires that distracts from the flow. 

 ○ Additional discussion of the mhealth landscape in which this project is situated may be more appropriate (which I know is included in the discussion, but could be alluded to earlier) 

 • Methods:

 ○ The design process is clearly explained, and reasonable. 

 ○ The figures are useful and give a clear sense of the structure and function of this application, however the authors could condense the description of the UX design process, focusing on key decisions and rationales rather than detailing every screen.

 ○ "To avoid mixing self-reported symptoms and more objective findings, 174 this question has been placed first in the examination."

 § This is very clever and an excellent idea. 

 ○ The use of the vibration function of a smart phone is also similarly brilliant, although with caveats as the authors note. 

 § I would add discussion of issues with variable vibration INTENSITIES in addition to the discussion of variable frequencies. 

 ○ The level of technical detail provided is excellent, however, I wonder if some of this detail is better suited for the supplemental portion. 

 • Results:

 ○ Expand this section significantly to provide more detailed analysis: 

 § Include a breakdown of scores for each subsection of the assessment (e.g., pin-prick, vibration, motor tests) comparing the Neuropathy Tracker results to the TNSc.

 § Provide statistical analysis for each subsection, not just the overall score.

 § Include a table showing mean scores, standard deviations, and p-values for each subsection.

 § Specifically address the validity of using the phone's vibration function compared to a tuning fork.

 ○ Clarify the statistical methods used, explaining what exactly is being compared in the t-test and why this test was chosen.

 ○ Consider adding a Bland-Altman plot --- or other detailed metric of the authors' choosing --- to visualize agreement between the two assessment methods.

 • Discussion / Limitations:

 ○ It is alluded to, but the authors should clearly highlight the limitation that. the small sample size likely limits some of the subsection by subsection analysis that would be necessary to fully validate this test.

 ○ In addition, the limitations section should highlight the lack of test-retest validation for this tool. 

 ○ Address accessibility features for users with visual or motor impairments.

 ○ Discuss what score threshold might be clinically meaningful for neuropathy detection or monitoring.

 ○ Address how the tool's performance might vary across different severities of neuropathy.

6. PLOS authors have the option to publish the peer review history of their article (what does this mean?). If published, this will include your full peer review and any attached files.

**Do you want your identity to be public for this peer review?** For information about this choice, including consent withdrawal, please see our Privacy Policy.

Reviewer #1: No

Reviewer #2: No

---

## [Decision Letter · Decision Letter 1]

13 Dec 2024

The Neuropathy Tracker – A mobile health application for ambulatory and self-administred assessment of neuropathy

PDIG-D-24-00371R1

Dear Prof. Bardram,

We are pleased to inform you that your manuscript 'The Neuropathy Tracker – A mobile health application for ambulatory and self-administred assessment of neuropathy' has been provisionally accepted for publication in PLOS Digital Health.

Best regards,

Haleh Ayatollahi

Section Editor

PLOS Digital Health

**Additional Editor Comments (if provided):**

**Reviewer Comments (if any, and for reference):**

Reviewer's Responses to Questions

**Comments to the Author**

1. If the authors have adequately addressed your comments raised in a previous round of review and you feel that this manuscript is now acceptable for publication, you may indicate that here to bypass the “Comments to the Author” section, enter your conflict of interest statement in the “Confidential to Editor” section, and submit your "Accept" recommendation.

Reviewer #1: All comments have been addressed

Reviewer #2: All comments have been addressed

2. Does this manuscript meet PLOS Digital Health’s publication criteria? Is the manuscript technically sound, and do the data support the conclusions? The manuscript must describe methodologically and ethically rigorous research with conclusions that are appropriately drawn based on the data presented.

Reviewer #1: Yes

Reviewer #2: Yes

3. Has the statistical analysis been performed appropriately and rigorously?

Reviewer #1: Yes

Reviewer #2: Yes

4. Have the authors made all data underlying the findings in their manuscript fully available (please refer to the Data Availability Statement at the start of the manuscript PDF file)?

Reviewer #1: Yes

Reviewer #2: Yes

5. Is the manuscript presented in an intelligible fashion and written in standard English?

Reviewer #1: Yes

Reviewer #2: Yes

6. Review Comments to the Author

Reviewer #1: The way the authors reframed the findings is appropriate and the quality of the manuscript has significantly improved. The study is promising, however still very preliminary.

Reviewer #2: • I appreciate the authors thoughtful response to the comments of myself and the other reviewers. I believe the current version of the manuscript has significantly clarified its scope and contribution. The paper is now more clearly reported as a feasibility / development study, and clearly lays out the direction for the further necessary evaluation.

• The methods section is more clear, and the design process has been more clearly explained. The further statistical analysis added is much more instructive.

I look forward to seeing the final validation study when it comes out

7. PLOS authors have the option to publish the peer review history of their article (what does this mean?). If published, this will include your full peer review and any attached files.

**Do you want your identity to be public for this peer review?** For information about this choice, including consent withdrawal, please see our Privacy Policy.

Reviewer #1: No

Reviewer #2: No
